# Tribology of Polymer Blends PBT + PTFE

**DOI:** 10.3390/ma14040997

**Published:** 2021-02-20

**Authors:** Constantin Georgescu, Lorena Deleanu, Larisa Chiper Titire, Alina Cantaragiu Ceoromila

**Affiliations:** 1Department of Mechanical Engineering, Faculty of Engineering, “Dunarea de Jos” University of Galati, 800008 Galati, Romania; larisa.chiper@ugal.ro; 2Department of Applied Sciences, Cross-Border Faculty, “Dunarea de Jos” University of Galati, 800008 Galati, Romania; alina.cantaragiu@ugal.ro

**Keywords:** polybutylene terephthalate (PBT), polytetrafluoroethylene (PTFE), blend PBT + PTFE, block-on-ring test, linear wear rate, friction coefficient

## Abstract

This paper presents results on tribological characteristics for polymer blends made of polybutylene terephthalate (PBT) and polytetrafluoroethylene (PTFE). This blend is relatively new in research as PBT has restricted processability because of its processing temperature near the degradation one. Tests were done block-on-ring tribotester, in dry regime, the variables being the PTFE concentration (0%, 5%, 10% and 15% wt) and the sliding regime parameters (load: 1, 2.5 and 5 N, the sliding speed: 0.25, 0.5 and 0.75 m/s, and the sliding distance: 2500, 5000 and 7500 m). Results are encouraging as PBT as neat polymer has very good tribological characteristics in terms of friction coefficient and wear rate. SEM investigation reveals a quite uniform dispersion of PTFE drops in the PBT matrix. Either considered a composite or a blend, the mixture PBT + 15% PTFE exhibits a very good tribological behavior, the resulting material gathering both stable and low friction coefficient and a linear wear rate lower than each component when tested under the same conditions.

## 1. Introduction

Due to a longer sequence of methyl groups in the monomer, the molecular chains of polybutylene terephthalate (PBT) are more flexible and less polar than those of polyethy1ene terephthalate (PET), inducing a lower melting temperature T (224–230 °C) and the glass transition temperature Tg (22–43 °C). This lower Tg allows for a fast crystallization when molding and shorter molding cycles with faster molding speed [1,2,3,4]. PBT, a semicrystalline engineering polymer, is included in the polyester class of resins and has a set of properties that recommends it in many special applications: rigidity and strength, combined with very good heat aging resistance. PBT-based materials (composites and blends) have better dimensional stability with uniform shrinkage behavior, stiffness, and heat resistance, low water absorption and high chemical resistance by incorporating fillers, reinforcing materials, and additives; material properties are tailored for user’s interest. They are processed mainly by injection molding [5]. PBT has restricted processability because of its processing temperature near the degradation one [6].

Lin and Schlarb [7] tested a hybrid material, short carbon fiber-filled PBT, with and without graphite as a solid lubricant, using a pin-on-disc tribotester, in dry regime. PBT with nanoparticles without graphite exhibits very good tribological performance, under moderate and severe load, with better mechanical characteristics as compared to the same material filled with graphite. Under 3 MPa and 2 m/s, the friction coefficient is 0.18 and wear rate is 0.8 × 10^−6^ mm^3^/Nm.

Dechet et al. [8] presented a laboratory technology for producing spherical polymer blend particles made of PBT and polycarbonate (PC) for selective laser sintering (SLS), including cogrinding and thermal rounding. The size distribution, shape and morphology of polymer constituents in this PBT + PC blend were analyzed.

Materials based on blends of PBT + PET with flame retardant agents (a new formulated agent, expandable graphite (EG), added separately or in a mixture of both) were tested for determining the influence of flame retardant [9]. Results demonstrate that incorporation of the mixture in PBT + PET blends could be recommended as potential applications for electronic household devices, products and automotive components, but research should be continued for their mechanical and tribological behavior.

In machine design, components made exclusively of polytetrafluoroethylene (PTFE) are rare, even if they maintain their properties at the initial values, independent of the manufacturing method [1], including chemical properties that remain unchanged for a long time (stability in chemically aggressive environments, insolubility, weather stability and anti-adhesion). The properties practically unaffected are flexibility at negative temperatures, thermal stability, a low dielectric constant and high resistance at electrical arch [10]. All experimental works underlined a very low friction coefficient [11,12] and high wear rate, especially in dry regime [13,14,15], but adding reinforcements as short fibers or powders reduces the wear by a factor of 100 or even more [13], especially in lubricated contacts [16,17]. Producers and users prefer to add PTFE in other polymers because of its poor wear resistance, being more efficient as solid lubricant [18] and rarely used as a matrix [19].

Polymer blends have been developed for their sets of characteristics, the mixture resulting in several new and improved properties or different from those characterizing each component alone [20]. The notion of a polymeric blend refers to materials artificially created, rationally combining different components to improve one or more characteristics and to diminish those that are not, based on theoretical models, laboratory tests and, finally, prototype results. At present, polymer alloys, polymer blends and their composites represent over 80% (by mass) of the total polymer-based materials [21].

In tribology, there are several polymer blends used for their good characteristics, especially using PTFE as a solid lubricant. Polymer blends are attracting the attention of specialists through a set of particular properties, such as low specific mass and strength-to-mass and stiffness-to-mass ratios that are superior to traditional materials, tribological properties [22], resistance to aggressive environments, electrical and thermal properties, which led to the use in the field of aeronautics, shipbuilding, electronics, medical components etc. Blends can be formed with miscible polymers, and homogeneous polymer mixtures up to the molecular level and with immiscible polymers, as in the PBT + PTFE blends.

Analyzing the properties presented in Figure 1, one may notice the narrow range for the melting temperature and the difference of about 100 °C for this characteristic. As PBT rapidly degrades above the melting temperature, it results in the blends of PBT + PTFE processed by mixing the melt PBT with solid PTFE powder; the dispersion quality depending on processing parameters [23].

As PTFE has a very low reactivity, when it is added in a harder polymer, as polyetheretherketone (PEEK), PBT or polyamide (PA), it exhibits an immiscible character, but, depending on the processing characteristics, the morphology of the blends obtained may vary from alternating microzones of PTFE and microzones of the other polymer to a fine dispersion of droplets.

Burris and Sawyer tested blends of PEEK + PTFE [25,26]. PEEK has good wear resistance and higher work temperature as compared to other thermoplastic polymers, a friction coefficient µ ~0.4 (dry regime) and a low thermal conductivity. Even if the recipes for polymeric blend with PTFE recommend 5–20% PTFE [18,27,28], Burris and Sawyer [25] reported that the polymeric blend with ~20% (vol) PEEK had a wear intensity 26 times lower as compared to that for PEEK, and 900 times lower than that exhibited by PTFE.

Briscoe and Sinha [19,29,30] made samples of PTFE with PEEK, using from 0% to 100% PEEK. Their results point out a monotonous increase of the wear rate and a monotonous decrease of the friction coefficient as the PTFE concentration increases. The differences between the results reported in [31] and [25,26] are related to material quality, their manufacturing process, microstructure and testing conditions. Bijwe et al. [31] tested abrasive resistance of PEEK + PTFE blends, with PTFE concentration up to 30% wt. Using a pin-on-disc tester, single pass condition against abrasive paper, low sliding velocity (v = 0.05 m/s), under loads of (6, 8, 10 and 12 N) and a very short sliding distance, L = 3.26 m, the wear rate was shown to increase with load and PTFE concentration.

These different results on blends of PEEK + PTFE underline the idea that friction and wear parameters fail to obey any mixture rule and laboratory tests; follow-up testing of actual systems is a necessity.

Research studies on tribological behavior of PBT as matrix with glass beads or short aramid fibers were reported in Georgescu et al. [32] and Botan et al. [33,34], only for a sliding distance of 5000 m.

An interesting tribological study was reported by Jozwik et al. [35], comparing several tribological characteristics for several polymeric materials, including PET + PTFE, PTFE + bronze and PTFE + graphite. Tests were done on a ball-on-disk system, with a sliding velocity of 0.8 m/s, the disk being made of polymeric material and the ball of aluminum oxide (Al_2_O_3_), for a sliding distance of 1000 m. The temperature was measured near the contact. The blend PET + PTFE (80/20) had the most convenient curve of temperature during the test, being characterized by a stable and low contact temperature, not exceeding 29 °C, under F = 30 N. The friction coefficient was 0.11 for F = 10 N and lower (0.07) for a higher load (F = 30 N). In addition, mass loss as a wear parameter was the smallest for a disk made of a PET + PTFE blend. Taking into account the similarity between the chemistry of PET and PBT, as members of the polyester polymers, the results presented in that study, even if the sliding distance seems too short for a comprehensive evaluation towards actual applications, is an inducement for testing PBT blends.

Research reports on the blends with both PBT and PTFE polymers are rare in literature, even if large polymer-producing companies [36] are using blends with PTFE for wear resistance applications.

A very important aspect in polymer blends is the nature of components, these could be miscible, partial miscible or even immiscible.

The main aim of this research is to emphasis the influence of PTFE concentration in PBT on tribological characteristics in dry regime.

## 2. Materials and Methods

The polymer (PBT) and the blends (PBT + PTFE) formulated for this tribological research were processed by die molding, resulting in bone specimens of type 1A, as recommended by SR EN ISO 527-2:2012 [37], at Monofil SA (Neamt, Romania). The commercial grade of PBT used was Crastin 6130 NC010 (as supplied by DuPont, Bucharest, Romania) [38].

The commercial grade of PTFE was NFF FT-1-1T^®^ Flontech (Ospitaletto, Brescia, Italy), with an average particle size of ~20 μm [39]. The dispersion of immiscible polymers is important in obtaining good results (dimension stability, mechanical and thermal, including impact, tribological characteristics, chemical resistance etc.).

For this study, the role of PTFE as an added material used in the following recipes was in concentrations of 5, 10 and 15% wt.

The parallelepiped block (dimensions of 16.5 mm × 10 mm × 4 mm) was obtained by cutting parts from the central bone specimens. The other component of the friction pair was the external ring of the rolling bearing KBS 30202 (Kedron Bearings Services LLC, Frankfort, KY, USA), having a diameter of Ø35 mm and a width of 10 mm, made of steel grade 100Cr6, with 60 −62 HRC and Ra = 0.8 μm. The shapes and dimensions of the friction couple (Timken-type) are presented in Figure 2.

The materials code and the average values for mechanical characteristics of the polymeric blends tested are presented in Table 1.

For the forces selected, the Hertz pressure in contact was calculated with the help of the Hertz contact calculator, included in UMT-2, UMT Test Viewer software (Version 2.14 Build 77, CETR, Campbell, CA, USA) [40], and the results are given in Table 2. Taking into account the elastoplastic characteristic of the block contact, these calculated values are estimated. A more realistic evaluation could be done with the help of the finite element method by introducing an adequate constitutive model of the block material. Values for Poisson coefficient were taken from literature and the values for the elasticity modulus (Young modulus) are average values experimentally determined by Georgescu from tensile tests. For all tests, the length of the contact between the block and steel ring was 4 mm.

The parameters of the block-on-ring test, for one material, are presented in Table 3. After a literature survey [7,41,42,43,44], each test was repeated twice, and the authors mentioned if the plots present one test or an average of the two tests.

The friction coefficient (COF) was monitored using a Universal UMT-2 (CETR^®^, Campbell, CA, USA) tribometer that had a transducer capable of measuring in actual time the friction force and calculating COF as a ratio between the normal force and the friction force, in any moment t of the test. The tribometer software [40] allows for viewing measured and calculated parameters.

The scanning electron microscope Quanta 200 3D from the Faculty of Mechanical Engineering (Technical University “Gheorghe Asachi” of Iasi, Iasi, Romania) and the scanning electron microscope FEI Quanta 200 (“Dunarea de Jos” University of Galati, Galati, Romania) were used for investigating the worn surfaces.

## 3. Results

Figure 3 presents the evolution of the friction coefficient for one test done for all investigated materials, L = 7500 m and F = 5 N, the longest test as concerning the sliding distance. At the lower velocity (v = 0.25 m/s), the neat polymer has a lower value of COF for a test, but the second test recorded higher values than those of the blends. For higher velocities, the blends PBT + PTFE have lower values, except for PF10 in the last third of the test. Short time oscillations of COF recorded for the blends may be generated by local disturbance in the components’ dispersions. It is important to mention that this stabilization of COF characterize forces F = 2.5 N and F = 5 N.

Investigation of the worn surfaces by scanning electron microscopy revealed deeper and fringier grooves, with larger rolled wear particles (see Figure 4a) as compared to the same aspects after testing at higher velocity, v = 0.75 m/s (see Figure 4b). For a higher load (F = 5 N), the aspect of worn surfaces is similar (see Figure 4c,d) with several deep traces, but there is no evidence of tearing-off a great volume of polymer as wear debris. In the scanning electron microscope (SEM) images of worn surfaces, the arrow indicates the sliding direction.

Analyzing the same worn surfaces at higher magnification (Figure 5), the lighter regime (F = 1 N, v = 0.25 m/s) produced worse damage (deeper grooves, larger wear debris). A higher velocity increases the temperature in the superficial layer and the cracks induced in the polymer are shorter and the wear debris are smaller and less numerous. This is an observation characterizing PBT sliding against steel. No analogy could be done with other polymers without investigations.

Adding PTFE in PBT, the aspect of worn surfaces changed (Figure 6). Noted were surfaces with local agglomeration, like that for PF5 and local zones depleted in PTFE as those for PF10 and PF15. The zones rich in PTFE alternate with depleted zones, meaning that the molding process could be modified for improving the dispersion. The flakes of PTFE embedded in the tribolayer seem to have lower size than 20 μm, meaning that the mixing procedure separate the initial particles from the smaller ones, which is beneficial for the tribological behavior.

PBT has average values of COF in the narrowest range, with greater average values for tests done with a sliding distance of L = 7500 m. The increase of average COF may result from the elimination of relatively larger wear debris, characteristic for this polymer. The average values of the friction coefficient are under 0.2 for tests under F = 5 N and all sliding velocities.

The average value of COF has a decreasing tendency for polymeric blends PBT + PTFE, at the sliding velocity of v = 0.75 m/s (Figure 7). At load F = 5 N, the polymeric blends have the friction coefficient lower for v = 0.5 m/s and v = 0.75 m/s, probably because at lower sliding velocity, PTFE is detaching in microribbons, especially when it is added in higher concentration (15%), ensuring, due to the lamination and the transfer of PTFE, a reduced friction. A similar wear process for PTFE and its composites are described by Gong et al. [45]. Jones et al. [46] pointed out high values of COF (over 0.6), for three polymeric balls sliding against a steel disk.

Figure 7 presents the average values obtained from two tests for the friction coefficient. Except for results obtained for the testing regime characterized by F = 1 N and v = 0.25 m/s, the value for COF is 0.15–0.18. This means that the tribological characteristic is less sensible to regime parameters (sliding velocity in the range of 0.5–0.75 m/s and load in the range F = 2.5–5 N), but also to PTFE concentration. This conclusion, based on the plots shown in Figure 7, means that investigation should extend the parameters toward higher loads, for the same materials. Adding PTFE does not change this parameter too much for the same regime ranges of inputs. As for the influence of sliding distance, COF is kept in a narrow range for L = 2500 m and with a larger spread for the longer distance, but the average values still remain in an acceptable range for practical use.

Myshkin et al. [47] presented trends of COF for polymeric materials sliding on steel, depending on sliding regime, including the plateau type, which is very advantageous for tribosystems functioning in dry conditions.

As the tribometer used for testing this class of polymeric materials is very accurate for measuring the linear wear, the authors calculated a linear wear rate of the block material, Wl, which gave the following relationship:(1)$$Wl=\frac{\Delta Z}{F\cdot L}\;\left(\mu m/(N\cdot km)\right)$$
where ΔZ (μm) is the approaching distance between the steel ring (considered rigid) and the block, recorded at the end of each test, F (N) is the normal force and L (m) is the sliding distance.

Mapping the tribological characteristics for parameters of interest as a function of a set of parameters is important in evaluating experimental results, as this mathematical modeling could reveal domains with optimum values for a certain set of testing parameters [48]. Maps allows for a panoramic view of the dependence of a tribological characteristic (here, the linear wear rate) on two variables [49]. The shape of a map could point out a change in wear mechanisms, a zone with optimal values for wear, even if tests were not done precisely for those indicated values.

Maps in Figure 8 and Figure 9 were drawn using a double spline technique using the software MATLAB R2009b (R2009b, MathWorks, Santa Clara, CA, USA) with each map representing the linear wear rate (Wl) as a function of the sliding velocity and PTFE concentration, with the help of a cubic interpolation. Map surfaces are “obliged” to pass through points given by experimental data (sliding velocity, PTFE concentration and linear wear rate). This methodology makes the map surface wavy, but the influence of one or a set of parameters is well evidenced. Comparing maps for different loads (Figure 8 for F = 1 N and F = 2.5 N), note that the greatest values are obtained for the lowest tested force (F = 1 N), meaning that there is an intense abrasive process (like a microcutting) as the superficial layer is not sufficiently compressed and the metallic counterpart (even with a high-quality texture) rasps the polymer and the polymer blends. This process is more intense for a high concentration of PTFE (15% wt) for low velocity (v = 0.25 m/s). It is interesting to note that for F = 1 N and L = 7500 m, the neat polymer has a greater linear wear rate, especially for high velocities (0.5 m/s and 0.75 m/s) meaning that wear processes were qualitatively modified. It is possible that high velocity makes the tribolayer soften and the polymer detachment becomes easier. The high linear wear rate for the blends with around 10% wt PTFE can be explained by several causes, including the existence of PTFE agglomeration that is torn off in larger microvolumes than those from blends having a better dispersion of PTFE in smaller microvolumes.

When the load increases from F = 1 N to F = 2.5 N, the linear wear rate decreases very much, underlining the idea that contact polymer (or polymer blend)–steel functions better when there is a sufficient load that does not allow for tearing off the polymer or the softer polymer, in the case of PBT + PTFE blends.

Figure 9 presents the maps of linear wear rate for the highest load F = 5 N. Note that the map scale becomes smaller when the sliding distance is increased. Analyzing Figure 9, the following conclusions may be formulated for each map: in Figure 9a, for L = 2500 m, the blends behave better for higher velocity (v = 0.75 m/s) and the lowest values for linear wear rate are obtained for PF5, but also for PF15. This could result from hard asperities and polymeric material. Obviously, droplets of PTFE are more rapidly and preferentially transferred. For Figure 9b, L = 5000 m/s, the best results were obtained for PBT at v = 0.25 m/s and for PF15 for all tested velocities. Values for PF10 are close to those obtained for shorter sliding distance. In Figure 9c, for the longest test (L = 7500 m), the map shape is similar to that for L = 2500 m but with lower values (almost three times lower for PBT at v = 0.75 m/s and two times lower for PF15). The blocks made of PF10 have the highest linear wear among the PBT + PTFE blends. Supplementary tests and investigations are needed for explaining this maximum or to check if a possible poor PTFE dispersion (the presence of agglomerates) could be the cause. The increase of linear wear rate for PF10 could be justified by nonuniform dispersion of PTFE, as revealed by SEM images. For these blocks, agglomeration of PTFE was found in the superficial layer, which was detached as larger wear debris.

## 4. Discussion on Tribological Processes

Wear mechanisms for polymeric materials have been discussed in Dasari et al. [50], Stachowiak and Batchelor [51] and Deleanu et al. [52], the main topics being abrasion, erosion, adhesion, transfer, fatigue, tribocorrosion and delamination as particular types associated with polymeric triboelements. The actual wear process is the result of synergic actions implying particular rubbing pair of materials and several wear mechanisms acting at the same time. Detailed description of these processes is given in Stachowiak and Batchelor [51].

Figure 10 shows evidence that wear mechanisms are present simultaneously on the worn surface of PBT. Each letter is written near the microzone where a certain wear mechanism is evident: A—fatigue cracks that are almost perpendicular to the sliding direction; B—abrasion trace with small depth, without rising edges and without material removal, typical for a polymer in normal regime; C—deep groove resulting from abrasive ploughing, also characterizing the polymer sliding against steel, with rising edges above the initial surface, repeatedly deformed; D—adhesion wear, which resulted from trapping and embedding of PBT wear debris, previously detached; and E—lateral lips and cracks, generated due to the viscoplastic nature of the polymer when hard asperities slide against the polymer.

Transfer films generated on polymer–metal rubbing contacts help for a gradual transition from transient to steady-state wear processes. The transfer mechanisms for PTFE and PTFE composites were explained and argued by experimental studies by Gong et al. [45] and Tomescu [13]. Gong et al. presents a model of adhesive wear for PTFE and PTFE composite with particles sliding against hard bodies (as those made of steel), in dry regime, with mechanical processes only, as PTFE is an almost inert material from the point of view of chemical reactions with the contacting materials (solids or lubricants). Tomescu presented images of adhesion and transfer of PTFE and PTFE composites when sliding in water against steel; the adhesion and transfer process is diminished very much for lubricating contacts and it preferentially involved PTFE and less for the harder material, when dealing with PTFE composites.

For PBT, the adhesion/transfer on steel counterbody has a lumpy character and the transferred debris are thicker and lumpy. For the blend PBT + PTFE, the transfer implies smaller debris of PBT, more PTFE and the agglomerations of wear debris occur in larger wear particles that are pressed in the steel surface texture. These attached agglomerations can explain the oscillations of friction coefficient, higher for low loads; when these particles are not sufficiently pressed and flattened, they become rolled (as shown in Figures 12c and 15c).

The abrasion of PBT blocks is evidenced by scratch traces of uneven depth and width, but with less evidence of detaching the polymer; this explains the very good tribological behavior of this polymer. Due to the viscous–plastic nature of the polymer, the grooves in the sliding direction generated by the metallic asperities have wavy edges, with lips due to the viscous flow and intermittent tears, with an oblique direction to sliding (Figure 11).

Wear debris shown in Figure 12 are different in shape and size: in Figure 12a, wear particle generated from the neat polymer that was trapped in a deep wear groove; in Figure 12b, two small agglomerations of PTFE particles that are very likely to be detached if the movement were to continue; in Figure 12c, a conglomerate of small wear debris made of PTFE, that adhered and bonded one to another, being pressed and rolled repeatedly in contact—the presence of such particles could be the cause of high oscillations of the friction coefficient, especially under lower loads; in Figure 12d, rolled wear debris with an extremely high concentration of PTFE; in Figure 12e, a droplet of deformed PTFE torn from its “bed” of PBT (up) and another round volume of PTFE, covered by a thin PBT bridge, probably formed by the wide spreading of a small volume of PBT; in Figure 12f, at higher sliding velocity, the wear debris made of PTFE have a butterfly aspect, but they are smaller and rolled.

Better dispersion was noticed for the block made of PF15 (Figure 13), tested at L = 7500 m. Worn surfaces are not gold coated before SEM investigation, but for even poor quality images, the dispersion can be seen, and the worn surface presents only small wear traces in depth and width.

The transfer on the metallic ring is very different, as revealed by comparing SEM images obtained after testing PBT, PBT + PTFE and PTFE (Figure 14): in Figure 14a, lumpy, refragmented wear debris deposit on the steel ring; in Figure 14b a rolled and pressed wear particle from the PTFE block, the folding of the this wear debris resulting from consecutive processes of laminating, adhering and rolling; in Figure 14c other wear debris from the block made of PTFE is pressed in the steel texture; this process was identified (less intense as thickness and area) for the PBT + PTFE blends.

PBT has a different transfer process (Figure 15): in Figure 15a wear debris are rare; in Figure 15b, wear debris transferred on the steel ring as lumpy islands, without being rolled. Figure 15c presents wear debris expelled from the contact near the friction path on the steel ring, when running a disk made of PF10; wear particles are made almost of PBT and these are robust, not rolled, shown in darker gray, and wear particles made almost of PTFE are white, thinner and rolled, many partially bonded to one another.

The blends PBT + PTFE have a transfer on the steel triboelement less intense as compared to that of neat PTFE (Figure 16a). The wear particles are smaller and not so agglomerated. The two particles in Figure 16b have different aspects; the bottom is more compact, very likely containing more PBT, with several microvolumes of PTFE (white). The other is intensely white, meaning its composition is consistent in PTFE. The aspect is rolled, and it is obvious that the agglomerated particle was generated by smaller adhering wear particles. The presence of more PTFE is revealed by the high degree of deformation. Figure 16c shows the details of two wear debris particles that could be considered extreme: the particle in the bottom of the SEM images is robust, thicker and not rolled (the gray shade characterizing PBT); the particle in the upper right corner of the same image is mostly made of PTFE, but also contains a small volume of PBT (gray shade). Both particles are conglomerates formed by the bonding and adhering of initially small wear debris.

## 5. Conclusions

This relatively new entry in the family of polymer blends, PBT + PTFE, is promising in tribological applications, at least for the parameters tested. By adding PTFE in PBT, the friction coefficient is kept in narrow range F = 2.5–5 N, v = 0.25–0.75 m/s and is less sensitive to PTFE concentration if the dispersion is of good quality. Local agglomerations of PTFE detach from the PBT matrix more easily and generate higher wear rates and oscillations of the friction coefficient.

Results for testing PBT + PTFE blends in dry conditions revealed that tribological characteristics are influenced by PTFE concentration and the test regime (load and sliding velocity). Components made of these blends can lower the power loss and enlarge the durability by reducing wear characteristics.

Linear wear rate have better values for longer sliding distance, meaning that wear is more intense at the beginning of sliding; the transfer process and the plastic deformation of the superficial layer allow for reducing friction and wear. The presence of PTFE reduces wear, especially for 5% and 15%. For 10%, this parameter increased, but the SEM investigation revealed poorer dispersion of PTFE. This decrease in wear rate is more obvious for higher velocities and loads, meaning that the polymeric material has to be compressed to make it less prone to be scratched and torn off by metallic texture.

In order to underline the better wear resistance of blends PBT + PTFE, the authors tested blocks made of neat PTFE for the sliding distance of L = 7500 m and load F = 5 N. When comparing to the results obtained for the blend PBT + 15% PTFE (PF15), it was in the favor of the blend:a.For v = 0.25 m/s, Wl_PTFE_ = 19.995 µm/(N∙km), approx. 70 times larger than that of PF15, (Wl_PF15_ = 0.283 µm/(N∙km));b.For v = 0.5 m/s, Wl_PTFE_ = 20.629 µm/(N∙km), approx. 85 times larger than that of PF15, (Wl_PF15_ = 0.240 µm/(N∙km));c.For v = 0.75 m/s, Wl_PTFE_ = 16.648 µm/(N∙km), approx. 77 times larger than that of PF15, (Wl_PF15_ = 0.216 µm/(N∙km)).

The experimental results, especially for the long sliding distance against steel (7.5 km), recommend using a PBT + 15% PTFE polymer blend as a replacement for components for tribological applications made only of PTFE in household appliances and automotive components. Components made of these blends could lower the power loss and enlarge the durability by reducing wear characteristics.

## Figures and Tables

**Figure 1 materials-14-00997-f001:**
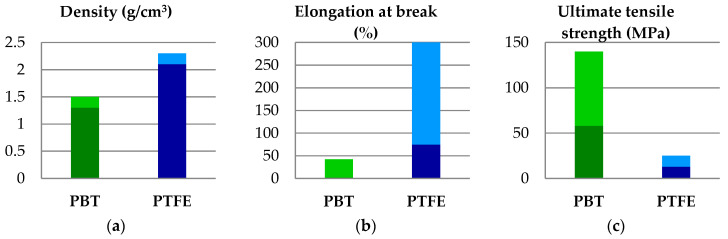
A comparison between characteristics for PBT and PTFE: (**a**) density; (**b**) elongation at break; (**c**) ultimate tensile strength; (**d**) flexural modulus; (**e**) melting temperature; (**f**) flexural strength. Mechanical characteristics are given for ambient temperature [24]. The dark color in each column represents the minimum value and the light color in each column is for the maximum value of the material characteristic.

**Figure 2 materials-14-00997-f002:**
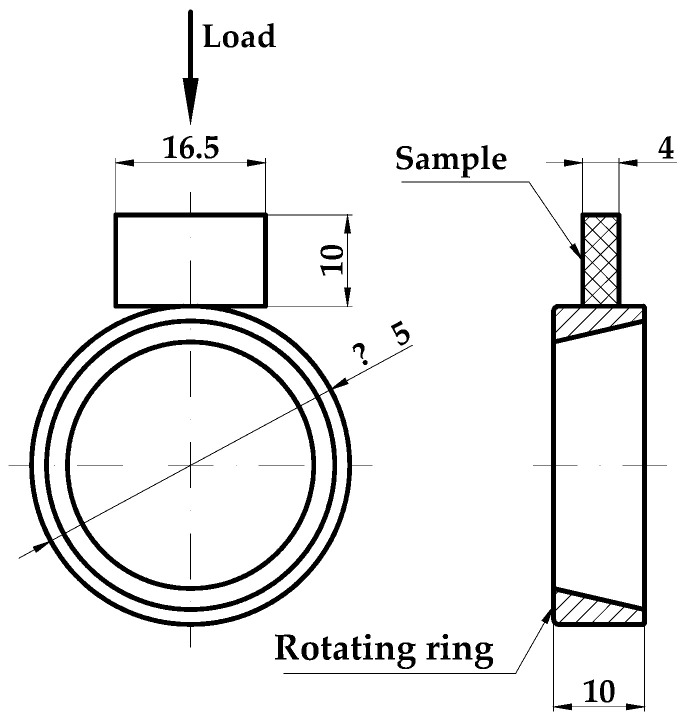
Dimensions (in mm) and shape of the tribotester elements.

**Figure 3 materials-14-00997-f003:**
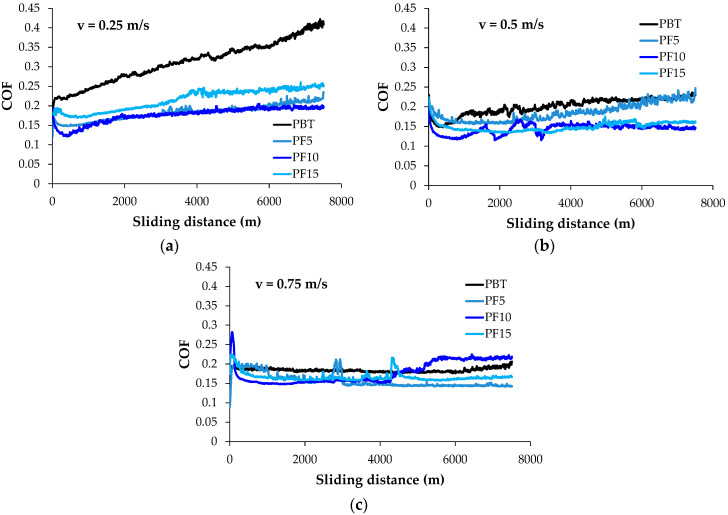
Evolution of friction coefficient (COF) in time for tested materials, F = 5 N, L = 7500 m: (**a**) v = 0.25 m/s; (**b**) v = 0.5 m/s; (**c**) v = 0.75 m/s.

**Figure 4 materials-14-00997-f004:**
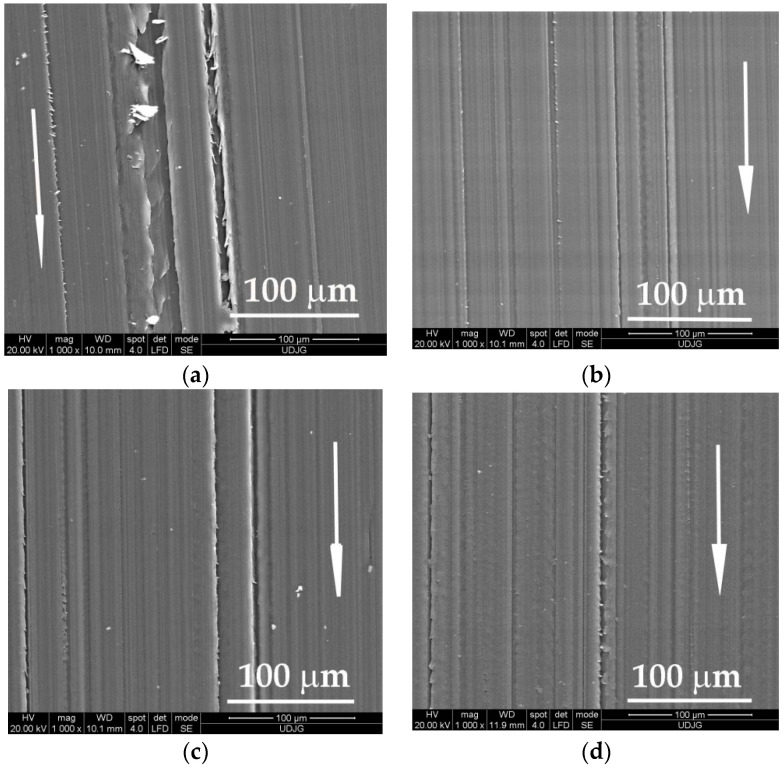
Appearance of worn surfaces for the block made of PBT, L = 5000 m: (**a**) F = 1 N, v = 0.25 m/s; (**b**) F = 1 N, v = 0.75 m/s; (**c**) F = 5 N, v = 0.25 m/s; (**d**) F = 5 N, v = 0.75 m/s.

**Figure 5 materials-14-00997-f005:**
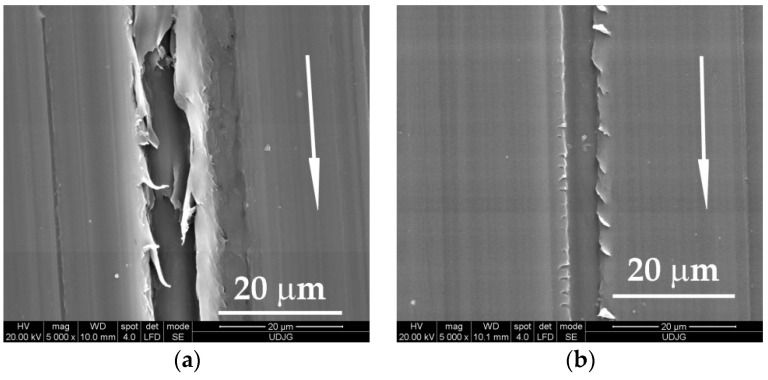
Details of the worn surface for the block made of PBT, L = 5000 m: (**a**) F = 1 N, v = 0.25 m/s; (**b**) F = 1 N, v = 0.75 m/s; (**c**) F = 5 N, v = 0.25 m/s; (**d**) F = 5 N, v = 0.75 m/s.

**Figure 6 materials-14-00997-f006:**
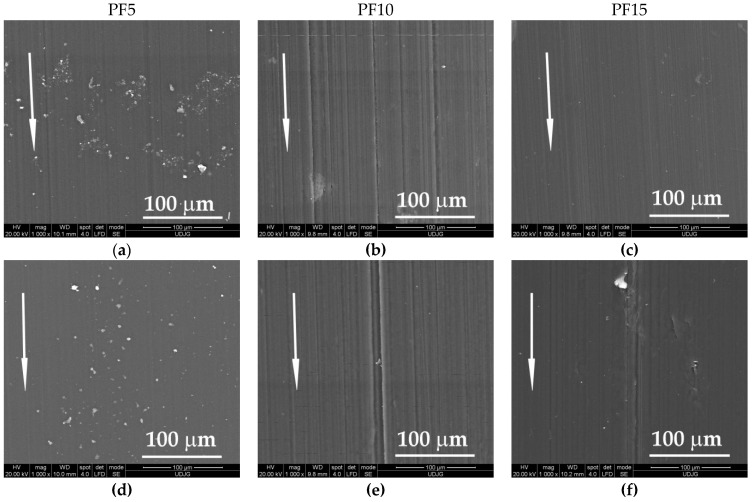
Details of the worn surface for the block made of PF5, PF10 and PF15, L = 5000 m and v = 0.25 m/s: (**a**) PF5, F = 1 N; (**b**) PF10, F = 1 N; (**c**) PF15, F = 1 N; (**d**) PF5, F = 5 N; (**e**) PF10, F = 5 N and (**f**) PF15, F = 5 N.

**Figure 7 materials-14-00997-f007:**
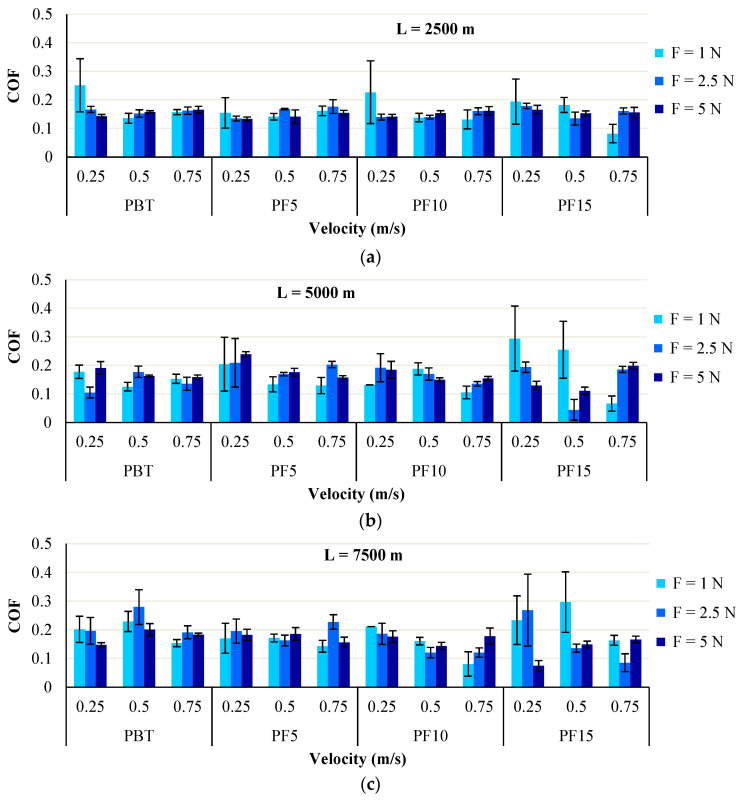
Average values of friction coefficient (COF), with standard deviation for tested materials, calculated for different sliding distances: (**a**) L = 2500 m; (**b**) L = 5000 m; (**c**) L = 7500 m.

**Figure 8 materials-14-00997-f008:**
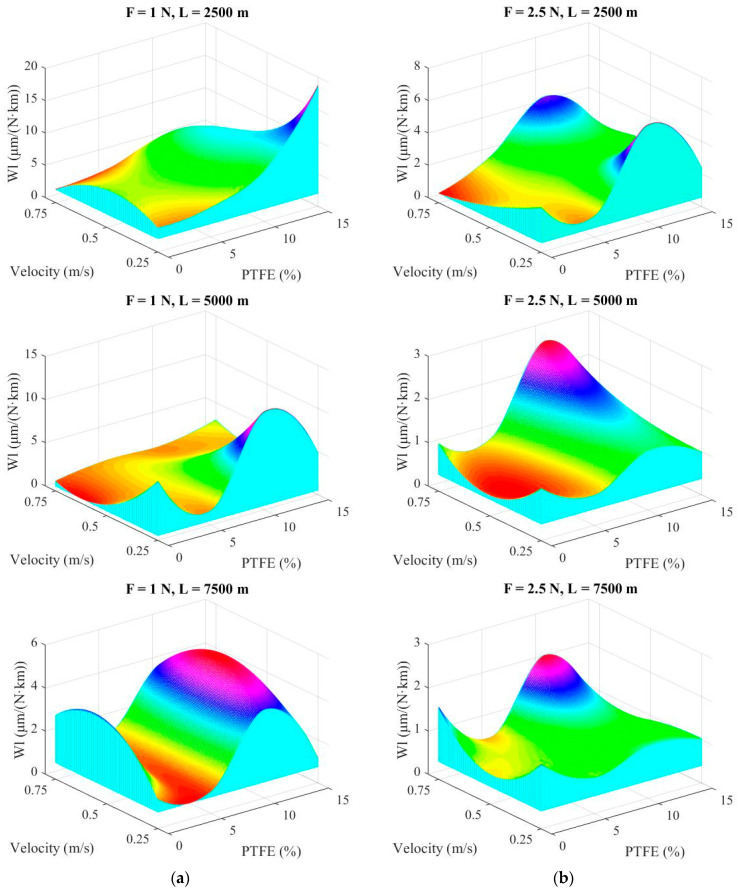
Maps of linear wear rate: (**a**) F = 1 N; (**b**) F = 2.5 N.

**Figure 9 materials-14-00997-f009:**
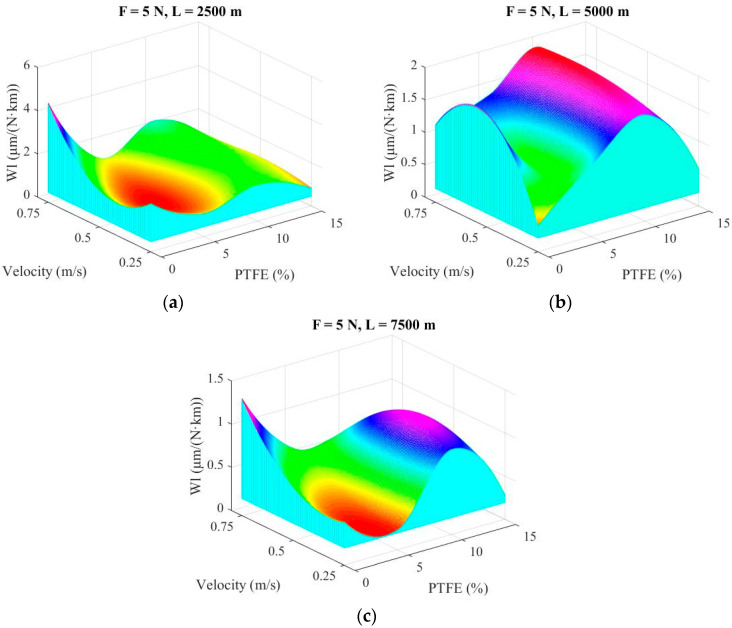
Maps of wear rate for tested load F = 5 N: (**a**) L = 2500 m; (**b**) L = 5000 m; (**c**) L = 7500 m.

**Figure 10 materials-14-00997-f010:**
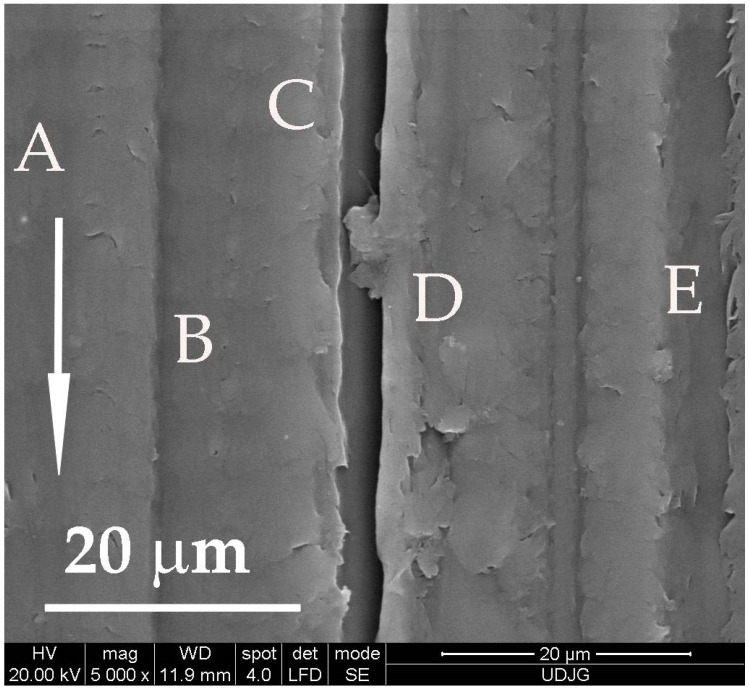
Wear mechanisms, identified on the worn surface of the block made of PBT (F = 5 N, v = 0.25 m/s, L = 5000 m).

**Figure 11 materials-14-00997-f011:**
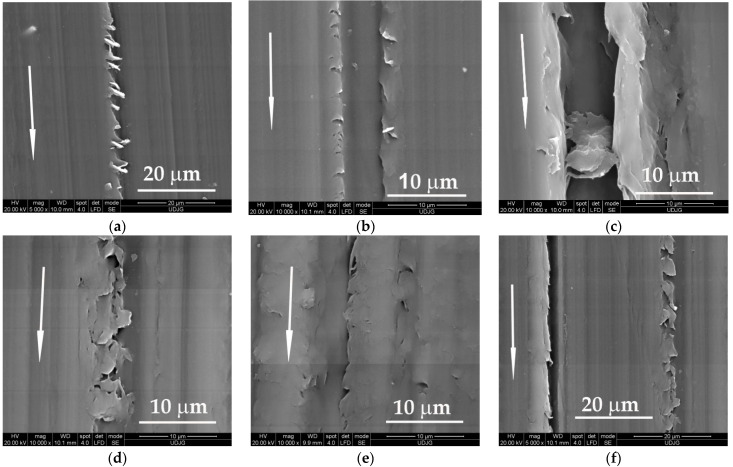
Wear mechanisms for PBT sliding against steel, L = 5000 m: (**a**) F = 1 N, v = 0.25 m/s; (**b**) F = 1 N, v = 0.75 m/s; (**c**) F = 1 N, v = 0.25 m/s; (**d**) F= 5 N, v = 0.25 m/s; (**e**) F = 5 N, v = 0.75 m/s; (**f**) F = 5 N, v = 0.25 m/s.

**Figure 12 materials-14-00997-f012:**
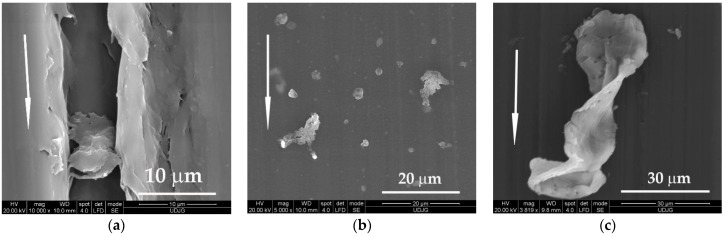
Wear debris after L = 5000 m: (**a**) PBT, F = 1 N, v = 0.25 m/s; (**b**) PF5, F = 5 N, v = 0,25 m/s; (**c**) PF10, F = 1 N, v = 0.25 m/s; (**d**) PF10, F = 1 N, v = 0.25 m/s; (**e**) PF15, F = 5 N, v = 0.25 m/s; (**f**) PF15, F = 5 N, v = 0.75 m/s.

**Figure 13 materials-14-00997-f013:**
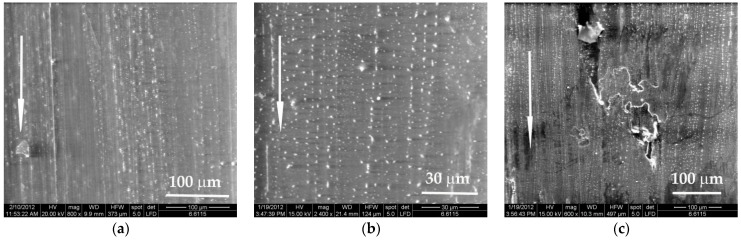
Aspects of worn surfaces for blocks made of PBT + 15% PTFE, tested under 5 N, for a sliding distance of L = 7500 m: (**a**) v = 0.5 m/s; (**b**) v = 0.25 m/s and (**c**) v = 0.25 m/s, worn surface with rolled conglomerate debris and thin debris reattached and pressed on the surface.

**Figure 14 materials-14-00997-f014:**
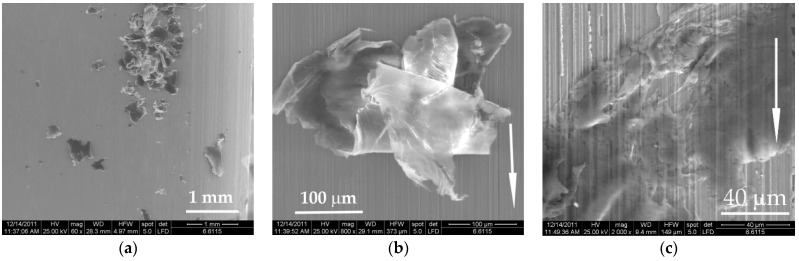
Transfer on the metallic disk from tests done in dry regime, on a block-on-ring tester, PTFE, tested under 5 N, v = 0.75 m/s, L = 7500: (**a**) lumpy, refragmentated wear deposit on the steel ring; (**b**) rolled and pressed wear particle from the PTFE block; (**c**) microzone with better adhered PTFE.

**Figure 15 materials-14-00997-f015:**
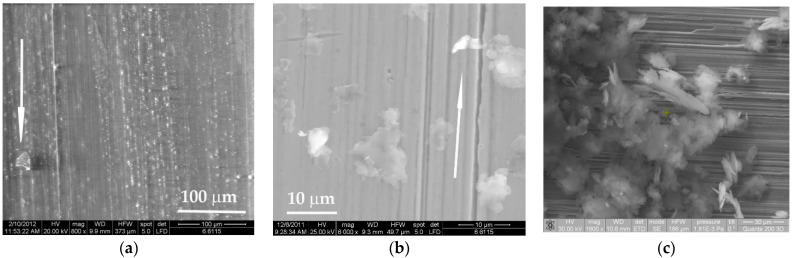
Worn surface of the PBT block, tested under 5 N, L = 7500: (**a**) abrasive wear and a wear particle reattached to the friction surface, v = 0.75 m/s; (**b**) v = 0.25 m/s abrasive wear located next to the friction path of the steel ring; (**c**) wear debris expelled from the contact near the friction path on the ring the block made of PF10.

**Figure 16 materials-14-00997-f016:**
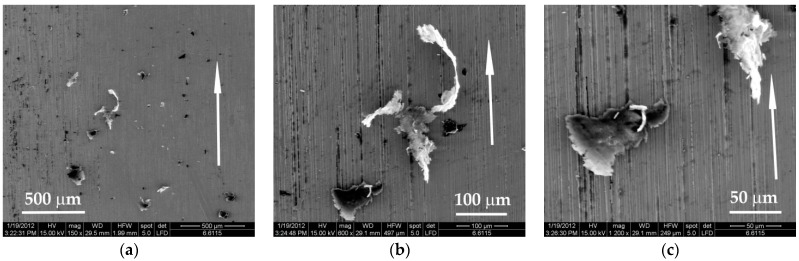
Wear debris from PBT + 15% PTFE blocks on steel counterparts, tested under 5 N, v = 0.25 m/s, L = 7500 m: (**a**) wear debris with different concentrations of PTFE (PTFE is bright white, PBT is gray); (**b**) two different wear particles; (**c**) detail of the particles in the previous image.

**Table 1 materials-14-00997-t001:** Average values for mechanical characteristics of the polymeric blends tested [23].

Material Code	Composition, % wt		Characteristic			
	PBT	PTFE	Young Modulus (MPa)	Ultimate Tensile Strength (MPa)	Elongation at Break (%)	Energy at Break (N∙m)
PBT	100	-	1923	41.5	9.4	17.6
PF5	95	5	1826	46.4	5.1	6.8
PF10	90	10	2202	36.9	2.4	1.8
PF15	85	15	1867	43.2	4.1	5.0

**Table 2 materials-14-00997-t002:** Hertz pressure for linear contact of the tested blocks made of PBT and PBT + PTFE blends.

Material Code	Hertz Pressure (MPa)		
	F = 1 N	F = 2.5 N	F = 5 N
PBT	3.1	5.0	7.1
PF5	3.1	4.9	7.0
PF10	3.4	5.3	7.6
PF15	3.1	4.9	7.0

**Table 3 materials-14-00997-t003:** Test parameters.

Normal Force (N)	Sliding Velocity (m/s)	Revolution Speed (rpm) *	Sliding Distance (m)		
			2500	5000	7500
			Testing Time		
1	0.25	136	2 h 46 min 40 s	5 h 33 min 20 s	8 h 20 min
	0.50	273	1 h 23 min 18 s	2 h 46 min 40 s	4 h 10 min
	0.75	409	55 min 33 s	1 h 51 min 7 s	2 h 46 min 40 s
2.5	0.25	136	2 h 46 min 40 s	5 h 33 min 20 s	8 h 20 min
	0.50	273	1 h 23 min 18 s	2 h 46 min 40 s	4 h 10 min
	0.75	409	55 min 33 s	1 h 51 min 7 s	2 h 46 min 40 s
5	0.25	136	2 h 46 min 40 s	5 h 33 min 20 s	8 h 20 min
	0.50	273	1 h 23 min 18 s	2 h 46 min 40 s	4 h 10 min
	0.75	409	55 min 33 s	1 h 51 min 7 s	2 h 46 min 40 s

* as an integer value in the test program.

## Data Availability

Data sharing not applicable.

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
