# Peer review of "Tribology of Polymer Blends PBT + PTFE"

_materials, 2021, doi:10.3390/ma14040997_

Round 1
Reviewer 1 Report
Georgescue et. al, report the tribological characteristics for PBT and PTFE blends. They analyzed friction coefficient and SEM images as a function of PTFE concentration. It is interesting but some Figures are difficult to understand. Fig. 2 shows the characteristics of PBT and PTFE, and it represents the density, elongation at break, etc with bar chart. Each bars are composed of two colors, for example, Fig. 2a density of PBT is composed of two green colors. What does it mean? Does it mean the error range?
Fig. 3 is the evolution of friction coefficient of samples, and Fig. 8 shows the average values of friction coefficient. Did you obtain the average values of COF from Fig. 3? As shown in Fig. 3, COF values are not constant and it changes. Therefore, it the average value should be compared with stand deviation. In Fig8, the standard deviation is missing. As shown in Fig. 3, the COF changes as sliding distance, and the average values should be provided with deviation.
In SEM images, I suggest to insert the sliding direction to understand the friction mechanism.
Fig. 9 of mapping using MATLAB is suspicious because the real data is only 3 and 4 points, i.e in case of velocity, the real measurements were done at 3 cases 0.25, 0.5 and 0.75. In case of PTFE concentration, the real measurements were done at 4 points, i.e 0, 5, 10, 15 %. In addition, the measurement values (may be the average values) did not include the deviation values, therefore, it is difficult to accept the result of cubic interpolation between points.
Finally, in the blend system, the dispersion is the most important factor to affecting mechanical properties. In this work, there’s no experimental results confirming the dispersion.
Some units and expressions were wrong. In Fig2, the unit of temperature was wrong. In table 1, the values of thermal expansion might be wrong, i.e -5 should be superscripts.
The maximum work temperature was written 110…180. Does it mean 110~180?
Please check the units and super/subscripts.
In page 2, it is written that “migration of EG locked the P-containing..” What is EG?
Author Response
Dear Reviewer,
Thank you for your comments. We did the modifications.

Reviewer 2 Report
This paper presents interesting results on tribology of PBT and PBT-PTFE polymer blends.
The paper can be read easily. It shows possible applications of the investigated materials.
I personally find the introduction section a bit long, very detailed. Not all the facts presented there seems to be relevant to this study. This paper has all in all 17 figures. I find this also a bit much, for only 17 pages of text. I also see there some potential of shortening, such as to present only Fig 6 and delete Fig. 5. Or combine them.
Page 5, second paragraph, last sentence: may the present tense of the verb is better here instead of past tense: ”the shapes… are presented…”
Table 4: why such not integer numbers are given for the revolution speed? Also, why so many decimals are reported for the testing time? In my opinion, it would have been better to give minutes and seconds here instead of using so many decimals.
Fig. 4. How reproducible were the results presented here? Were all the time the curves with that small differences in-between?
The scale bar of SEM images (Figs. 5-7 and 12-17) are hard to read in my copy. Maybe the authors could add an extra scale bar here?
First paragraph on page 14, which seems to be the same as the last paragraph before conclusion section: maybe the authors could keep only one of these paragraphs and move it to acknowledgment or experimental section. As it is now, in the middle of the manuscript, looks a bit strange to me.
Second paragraph on page 14: what were the conclusions of these two papers? Can the authors shortly describe/ mention that again?
Author Response

(The authors gave the same response as above.)

Reviewer 3 Report
Comments for authors are in the attachment.

Author Response

(The authors gave the same response as above.)

Reviewer 4 Report
The aim of this paper is mainly to present results on tribological characteristics (friction and wear) for polymer blends made of polybutylene terephthalate (PBT) and polytetrafluoroethylene (PTFE). Even if the topic is interesting I have some deep concerns from a tribological point of view to be addressed before consider this research publishable in a quality journal. My main concerns are:
1) Did the Authors considered the requirements of the ASTM G77 standard for tribological investigation executed by Block-on-Ring tester?
2) What is the contact pressure reached during the tests?
3) The Authors declared to measure COF by using Tribometer - CETR - UMT-2. At my best knowledge this tribometer enables two modes of sliding contacts: reciprocating or unidirectional sliding in environmental temperature and controlled air humidity. What were the friction tests conditions? What is the connection between the Block-on-Ring tester used bro measure wear rate and CETR UMT2 to measure friction forces?
4) COFs values must be presented accounting for their variance
5) The Authors declared to perform twice each test. Is this condition usually recognized from the scientific community? Usually at least three tests are performed...
6) Did the Authors considered the contact temperature variations during the tests? For such kind of materials the connection between temperature and wear is a key factor....
7) Figures 9 and 10 need a deeper theoretical explanation, also, in the framework of available scientific literature..
8) the conclusions must take into account the above considerations and must underline the limitations of this research.
Author Response

(The authors gave the same response as above.)

Round 2
Reviewer 1 Report
All answers for the questions are satisfactory, and they revised the manuscript as the reviewer's comments. Therefore, I suggest "accept as is".
Author Response
Thank you for your comments.
Reviewer 3 Report
Accept in present form.
Author Response
Thank you for your comments.
Reviewer 4 Report
The Revision made by the authors is not very deep. However, the Authors addressed most of my concerns. Unfortunately my main issue in 3) "What is the connection between the Block-on-Ring tester used bro measure wear rate and CETR UMT2 to measure friction forces?" was totally ignored.
This point is for me a key point of this investigation: certainly the Authors, as tribologists, know that "...friction is a characteristic of the tribosystem and not a fundamental material property. Therefore, a given set of materials can rank in a different order when friction is tested in different kinds of tribometers." (P.J.Blau)
Moreover the answer to issue 4) "We added the extreme values (maximum and minimum ones) for the COF in the graphs." is not an usual procedure in the presentation of COFs results. Please add variance to each proposed COF value.
Author Response
Thank you for your comments.

Round 3
Reviewer 4 Report
The Authors addressed my concerns.
Author Response
Thank you for your comments. They were useful to make the paper better.
